# Antimicrobial Resistance Genes Analysis of Publicly Available *Staphylococcus aureus* Genomes

**DOI:** 10.3390/antibiotics11111632

**Published:** 2022-11-16

**Authors:** Vincenzo Pennone, Miguel Prieto, Avelino Álvarez-Ordóñez, José F. Cobo-Diaz

**Affiliations:** 1Department of Food and Drug, Università degli Studi di Parma, 43121 Parma, Italy; 2Department of Food Hygiene and Technology, Universidad de León, 24071 León, Spain; 3Institute of Food Science and Technology, Universidad de León, 24071 León, Spain

**Keywords:** antimicrobial resistance, whole genome sequencing, *Staphylococcus aureus*, surveillance, food safety

## Abstract

*Staphylococcus aureus* is a pathogen that can cause severe illness and express resistance to multiple antimicrobial agents. It is part of the ESKAPE organisms and it has been included by the Centers for Disease Control and Prevention (CDC) of USA in the list of serious threats to humans. Many antimicrobial mechanisms have been identified, and, in particular, antimicrobial resistance genes (ARGs) can be determined by whole genome sequencing. Mobile genetic elements (MGEs) can determine the spread of these ARGs between strains and species and can be identified with bioinformatic analyses. The scope of this work was to analyse publicly available genomes of *S. aureus* to characterise the occurrence of ARGs present in chromosomes and plasmids in relation to their geographical distribution, isolation sources, clonal complexes, and changes over time. The results showed that from a total of 29,679 *S. aureus* genomes, 24,765 chromosomes containing 73 different ARGs, and 21,006 plasmidic contigs containing 47 different ARGs were identified. The most abundant ARG in chromosomes was *mecA* (84%), while *blaZ* was the most abundant in plasmidic contigs (30%), although it was also abundant in chromosomes (42%). A total of 13 clonal complexes were assigned and differences in ARGs and CC distribution were highlighted among continents. Temporal changes during the past 20 years (from 2001 to 2020) showed that, in plasmids, MRSA and macrolide resistance occurrence decreased, while the occurrence of ARGs associated with aminoglycosides resistance increased. Despite the lack of metadata information in around half of the genomes analysed, the results obtained enable an in-depth analysis of the distribution of ARGs and MGEs throughout different categories to be undertaken through the design and implementation of a relatively simple pipeline, which can be also applied in future works with other pathogens, for surveillance and screening purposes.

## 1. Introduction

*Staphylococcus aureus* is a bacterial pathogen that can cause cases of serious bacteraemia and other infections, such as endocarditis, prosthetic device infections (PDI), pleuropulmonary infections, abscesses, meningitis and urinary tract infections (UTIs), in humans [1]. It is also a major concern in the food production industry because some clonal lineages of this pathogen can be transferred from animals to humans, with possible associated health issues and economical losses [1,2,3]. Its capacity for adaptation and resistance to antimicrobials makes it a very dangerous microorganism. Indeed, it is included in the list of serious threats to humans by the Centers for Disease Control and Prevention (CDC) of USA and in the World Health Organisation (WHO) list of high priority pathogens for the development of new antibiotics [4,5]. Antimicrobial resistance (AMR) is a serious issue in both clinical and food production environments due to the possible spread of superbugs with multi-drug resistance (MDR) mechanisms [6]. The acquisition of AMR by *S. aureus* strains started with the extensive use of penicillin in the 1940s and the use of methicillin later, leading to the development of methicillin-resistant *Staphylococcus aureus* (MRSA) [7]. As a result, *S. aureus* has been included in the group of ESKAPE pathogens, a group of MDR organisms of major concern for humans [8].

Whole genome sequencing (WGS) is nowadays a commonly used and powerful tool for the characterisation of pathogenic bacteria, including *S. aureus* [9]. It enables the typing of strains (e.g., using core genome multi locus sequence typing), as well as the prediction, with a certain level of confidence, of their phenotypic characteristics, such as virulence and AMR potential, by searching for the genes which determine these phenotypic traits in the genomes or specifically in their mobile genetic elements (MGE) [10]. Some studies have found a high level of confidence between AMR phenotypes and genotypic predictions, although sometimes the correlations obtained are not absolute, meaning that the standard microbiological protocols for AMR assessment still need to be used [11,12]. WGS is also being used for outbreak investigations because it facilitates the fast discovery of the source of outbreaks [13]. Thanks to the diffusion of open science practices, WGS research data are being made freely available in public databases. When corresponding metadata are also incorporated, such as information on isolation source, country, time of isolation, etc., the investigation of trends in distribution of genetic determinants as associated with particular origins is facilitated. In this manner, the aim of this study was to provide an overview of the antimicrobial resistance genes (ARGs) of *S. aureus* in publicly available genomes, with a focus on their distribution through continents, isolation sources, clonal complexes (CCs), and the temporal changes along the past 20 years. The pipeline used in this study can be easily applied for further analyses, for example considering other pathogens, or for surveillance purposes, such as to monitor changes in ARGs distribution in source categories and countries, in relation to the antibiotic usage.

## 2. Results

A total of 29,679 *Staphylococcus aureus* genomes were downloaded from the three public databases. Of these, 24,765 chromosomes containing ARGs were detected. A total of 1,063,861 plasmidic contigs were identified, with 21,006 ARG-containing plasmidic contigs. In total, 13 CCs were identified, with CC8 (29%), CC22 (21%), and CC5 (19%) as the most abundant for chromosomes, and CC8 (40%), CC5 (15%), CC22 (12%), and CC1 (9%) for plasmids. Some of the WGS data could not be assigned to any known CC (8% of chromosomic and 6% of plasmidic contigs). Information on the isolation source was not available for 52% of the chromosomic and 86% of the plasmidic contigs (Appendix A, metadata information per chromosomic or plasmidic contig can be found at Appendix A). The most abundant isolation sources were human (37%), other (6%), animal (2%), environment (1.6%), and food (1%) for chromosomic data, and human (11%), animal (1.6%), other (0.7%), food (0.5%), and environment (0.2%) for plasmidic contigs. The source “other” includes all the isolation sources that could not be assigned to the rest of source categories. More than half of the chromosomic contigs belonged to isolates from unknown countries (57%), while 19% were from North America, 13% from Europe, and 6.4% from Asia. Regarding the plasmidic contigs, 85% belonged to isolates from unknown countries, 6.5% from Europe, 2.3% from North America, 4.3% from Asia, 0.7% from Oceania, 0.7% from Africa, and 0.3% from South America. The distribution of isolates from different isolation sources was different in different continents (Appendix A). For example, *S. aureus* chromosomes from human isolates were obtained mostly from North America and Europe (47% and 28%, respectively), while most of food chromosomes were from Asia and Europe (54% and 34%, respectively), most of animal chromosomes were from Europe and Asia (47% and 29%, respectively), and most of the chromosomes from environmental samples were from Asia (29%) and North America (28%). Interestingly, plasmidic contigs showed a different distribution between isolation sources and continents. For example, most plasmidic contigs from human isolates originated from Europe and Asia (50% and 21%, respectively), while animal and food plasmidic contigs were mostly from Asia (77% and 82%, respectively), and the environmental ones mostly from Europe (84%). All the raw data regarding genome codes, collection year, country, host, and MLST information for *S. aureus* chromosomic and plasmidic contigs are included in the Appendix A, respectively.

A total of 73 different ARGs were identified in chromosomes and 47 in plasmidic contigs. Globally, the data analysis showed that 95% of chromosomes harboured at least one ARG associated with beta-lactam resistance, followed by aminoglycoside (54%), macrolide (30%), and tetracycline (24%), while 44% of plasmidic contigs were carrying ARGs for macrolide resistance, followed by beta-lactam (31%), aminoglycoside (28%), and tetracycline (15%) (Figure 1). The correspondence between ARGs and antibiotic families is available in the Appendix A.

The most abundant ARGs harboured on chromosomes were *mecA* in 84% of chromosomes, *blaZ* (42%), *ant(6)-Ia* (28%), *aph(3’)-III* (24%), *aadD* (21%), *tet(M)* (20%), *spc* (18%), and *erm(A)* (17%) (Figure 1). The plasmidic contigs harboured mostly *blaZ* (30%), *spc* (20%), *erm(A)* (19%), *erm(C)* (15%), and *tet(K)* (13%) (Figure 1).

Co-occurrence of ARGs was also assessed, and 1041 different genotypes were identified in chromosomes, with *mecA* (24%), *blaZ* (6%), *mecA-ant(6)-Ia-aph(3’)-III* (5%), *mecA-blaZ-ant(6)-Ia-aph(3’)-III-aac(6’)-aph(2’’)-tet(M)-dfrG* (4%), *mecA-blaZ-aadD-spc-erm(A)* (4%), and *mecA-blaZ* (3%) as the most abundant ones (Table 1 and Figure 1). Only 100 different genotypes were identified in plasmidic contigs, and the most abundant were *blaZ* (28%), *spc-erm(A)* (19%), *erm(C)* (15%), *tet(K)* (13%), and *msr(A)-mph(C)* (7%) (Table 1 and Figure 1).

### 2.1. Geographical Distribution of CCs and ARGs

The geographical distribution of *S. aureus* CCs showed differences by continent. In the chromosomes, 56 and 47% of CC5 were present in North America and South America, respectively, while 32 and 23% of chromosomic CC22 and CC398 were identified in Europe, respectively, and CC8 in Africa, Asia, and Oceania (28%, 25%, and 23%, respectively) (Figure 2A). Within continents, variable patterns were observed among different countries. For example, in North and South America, CC5 and CC8 were the most abundant in USA, Argentina, Brazil, and Colombia (CC5: 56%, 28%, 53%, and 53%, respectively; CC8: 33%, 19%, 11%, and 31%, respectively), however, CC30 was also abundant in Argentina (20%). Likewise, 13, 36, 21, and 90% of chromosomic CC398 were from Denmark, Germany, Italy, and Netherlands, respectively, while this CC was absent in UK, where CC22 was the most abundant CC (22%). Other dominant CCs were CC22 (10%), CC5 (12%), and CC8 (24%) in Denmark; CC22 (15%) and CC5 (11%) in Germany; and CC5 (23%) and CC1 (10%) in Italy. CC59 and CC1 were the most abundant CCs (20% and 23%, respectively) in China; CC8 (57%) and CC22 (37%) were predominant in Singapore; CC45 (33%) and CC8 (33%) in Taiwan; and CC121 (39%) and CC8 (13%) in Thailand (Figure 2A).

Differences in the CCs distribution were also observed for plasmidic contigs (Figure 2B). Plasmidic contigs linked to CC8 were quite abundant throughout all the continents, with the exception of Oceania (Africa, 20%; Asia, 29%; Europe, 9%; North America, 25%; and South America, 11%). CC5 plasmidic contigs were abundant in North America (49%), South America (53%), Europe (12%), and Africa (13%). Plasmidic contigs from CC22 were also abundant in Europe (31%), while plasmidic contigs from CC1 were abundant in Asia (26%), Oceania (54%), and South America (10%). Remarkably, 30 and 46% of plasmidic contigs belonged to unassigned CCs in Africa and Oceania, respectively. A few differences among countries from the same continent were also found for plasmidic contigs. CC398 and CC45 plasmidic contigs were the most abundant in Germany (26% and 20%, respectively), where 28% of plasmidic contigs were from unassigned CCs, while CC1 and CC59 plasmidic contigs were quite abundant in UK (10% and 11%, respectively). Those from CC22 were abundant in Singapore (26%), and from CC45 in Taiwan (74%), despite both being absent in China, where 58% of plasmidic contigs belonged to CC1 and 34% to unassigned CCs (Figure 2B).

Significant differences (*p* < 0.05) were observed in the distribution of chromosomic ARGs among Continents (Figure 3A). For example, *aac(6′)-aph(2″)* and *catpC221* were found significantly more abundant in Asia (44 and 3% of chromosomes from Asia, respectively) and in South America (38% and 3%, respectively) than in Europe (15% and 0.1%, respectively), even though *catpC221* occurrence was higher in Africa, but with no significant differences if compared to its occurrence in other continents (*p* > 0.05); *tet(K)* and *tet(M)* were more represented in Europe (15% and 27%, respectively) than in South America (6%, in both cases); *aadD* had very small presence in Africa (0.3%), while it was much more prevalent in North America (53%) than in South America (12%), Europe (9%), or Asia (21%). The ARGs *ant(6)-Ia* and *aph(3’)-III* were significantly more abundant in South America (46% in both cases) than in all the other continents, excluded North America (32% and 6%, respectively) and *dfrG* was found more abundant in Africa (39%) and Asia (32%) than in Europe (6%) and North America (1%) (Figure 3A).

The distribution of plasmidic ARGs throughout the continents showed statistically significant differences for only four genes: *erm(C)* was found more abundant in plasmidic contigs from Asia (24%) than from North and South America together (7%), while *mph(C)*, *msr(A),* and *spc* were more abundant in North/South America (15%, 15%, and 22%, respectively) than in Europe (3%, 4%, and 4%, respectively) (Figure 3B).

### 2.2. CCs and ARGs Distribution by Source of Isolation

In chromosomes, 35.9% of animal sources belonged to CC398 and 34.56% of human sources to CC5, followed by CC1 in animal (24.36%) and CC8 in human (24.49%) sources (Figure 4A). CC398 was also found in abundance in the categories Environment and Food (21.41% and 17.05%, respectively), while CC8 was prevalent in the Environment category (15.11%), however, these differences were not significant (*p* > 0.05).

The analysis of plasmidic contigs showed a predominance of CC1 plasmidic contigs in isolates from animal sources (73%) and those from CC398 in environmental isolates (74%) (Figure 4B). Furthermore, plasmidic contigs from human isolates seemed to be more represented in CC8 (21%), CC22 (22%), and CC5 (20%) (Figure 4B). Additionally, in this case, the statistical analysis showed no significant differences.

Chromosomal *mecA* and *blaZ* were widespread throughout the isolation sources, although *mecA* was significantly (*p* < 0.05) more abundant in human (84% of human isolates were carrying *mecA*) than in animal isolates (74%), while *blaZ* was significantly more abundant in isolates obtained from environmental samples (70%) than from animals (32%) (Figure 5A). Some other genes showed differences among isolation sources. Thus, *aac(6′)-aph(2″)* and *tet(M)* were more abundant in the animal source (42% and 45%, respectively) than in all the other sources, although these differences were not significant, apart from *aac(6′)-aph(2″)* in human isolates vs. other isolates (16% and 8%, respectively) (Figure 5A). Eventually, 13 and 3% of environment and animal isolates, respectively, were carrying *aph(3′)-III*, and 15 and 3% of environment and animal isolates, respectively, were carrying *dfrK* (*p* < 0.05, Figure 5A).

The ARGs found in plasmidic contigs from different sources showed a high abundance of *tet(K)* in the environment (60%) and *blaZ* throughout the different isolation sources, with no significant differences (*p* > 0.05, Figure 5B).

### 2.3. ARGs Distribution among S. aureus CCs

Almost all the CCs were carrying *mecA*, although with less abundance in CC15 (9.8% of CC15 isolates were carrying *mecA*) and CC121 (4.03%), while it was absent in CC130. *mecA* was significantly more abundant in CC8 (94.76%), CC22 (98.46%), and CC5 (92.77%) than in CC1 (74.46%) (Figure 6A). *blaZ* was significantly more abundant in CC8 (67.77%) and CC5 (45.30%) than in CC22 (0.21%), while in CC30 (88.87%) and CC8 it was more abundant than in CC5 (Figure 6A). *tet(K)* seemed to be more abundant in CC121 (70.16%), however, significant differences (*p* < 0.05) were detected only between CC5 (1.39%), CC59 (31.69%), and CC8 (3.27%) (Figure 6A). Other significant differences were found for *erm(B)*, significantly more abundant in CC59 (43.66%) than in CC5 (0.29%) and CC8 (0.34%), and in CC398 (4.21%) than in CC5 and CC8; *erm(A)*, more dominant in CC5 (62.96%) than in CC22 (0.04%) and CC8 (8.19%); *aph(3′)-III*, more abundant in CC8 (65.93%), but only when compared to CC22 (0.06%) (*p* < 0.05); *aac(6’)-aph(2″)*, with higher occurrence in CC8 (47%) than in CC5 (13.15%) and CC59 (2.46%); *aadD*, more present in CC5 (59.52%), but only when compared to CC8 (17.30%) (Figure 6A); *ant(6)-Ia*, more abundant in CC8 (65.25%), but only if compared to CC30 (0.67%) and CC22 (0.00%); *tet(M)*, predominant in CC8 and CC398 (83.56% and 48.71%, respectively), with significant differences when compared to almost all the other CCs (Figure 6A); and *tet(L)*, more abundant in CC398 (11.68%) than CC5 (0.84%). All the frequencies of ARGs in the different CCs are listed in the Appendix A.

Some ARGs were significantly more abundant in plasmidic contigs belonging to particular CCs. For example, *blaZ* was more abundant in plasmidic contigs from CC45 (73% of CC45 isolates were carrying *blaZ*) and CC5 (41%) than from CC8 (20%); it was also more abundant in CC1 (52%) than CC22 (3%) and CC8 (Figure 6B). The presence of *erm(C)* was significantly higher in plasmidic contigs from CC1 (11%) than from CC5 (6%); and in CC22 (78%) than in CC1, CC5, and CC8 (3%) (Figure 6B). Furthermore, *tet(K)* presence in plasmidic contigs from most CCs was significantly higher than in CC5 (1%), excluded CC93 (0.2%), and *aadD* occurrence was higher in CC5 (12%) and CC22 (11%) than CC1 (1%) (Figure 6B and Appendix A).

Some characteristic patterns of association of CCs with ARGs linked to resistance to particular antimicrobial families were observed (Figure 7). Globally, all the chromosomes analysed were carrying ARGs associated with resistance to Beta-Lactams and Aminoglycosides, with chromosomes belonging to CC8 and CC5 carrying ARGs associated with resistance to a maximum of 13 different antimicrobial families, while CC130 only to 2 (Appendix A). All the plasmidic contigs analysed carried at least one ARG associated with resistance to the macrolides, aminoglycosides, and tetracyclines classes, with plasmidic contigs belonging to CC8 and CC5 carrying ARGs associated with resistance to a maximum of 11 antimicrobial families, while CC130 and CC121 only to 4 (Appendix A).

In chromosomes, some common patterns of (co)occurrence were observed in particular CCs (Figure 7A). For example, the occurrence of beta-lactams ARGs was lower in CC121 (11%) than in other CCs, while tetracyclines and trimethoprim ARGs were more abundant in CC398 (85% and 38%, respectively) and CC121 (70% and 51%, respectively) than in other CCs (Figure 7A). Another cluster, formed by four CCs (CC8, CC1, CC5, and CC59) showed a significantly higher occurrence of aminoglycosides ARGs (82%, 70%, 81%, and 48%, respectively), while CC1, CC8, CC5, and CC398 showed significantly higher abundance of macrolides ARGs (20%, 27%, 65%, and 30%, respectively) compared to CC121 (13%) (Figure 7A).

In plasmidic contigs, CC398 and CC121 clustered together, with high occurrence of tetracyclines ARGs (45% and 67%, respectively) (Figure 7B). Despite the many diversities observed in the occurrence of ARG families among the CCs, significant differences were found only for plasmidic contigs from CC1 and CC45, with a higher occurrence of aminoglycosides ARGs than CC93 (5%, 7% and 1%, respectively) (Figure 7B).

### 2.4. Temporal Changes in ARGs

From the whole dataset, information regarding the isolation dates (from 2001 to 2020) were present only in 10,927 chromosomes (44%) and 2855 plasmidic contigs (14%), which were employed for the following analysis. The analysis of temporal changes in ARGs present on chromosomes showed that over the last 20 years the global occurrence of ARGs has changed, in some cases showing significant differences (*p* < 0.05, Figure 8, Appendix A). For example, *mecA* presence has declined significantly up to 2010, with no significant differences after that, while the reduction in *blaZ* presence in the past five years is not statistically significant compared to the previous years (Figure 8). A reduction was also observed in other genes, such as *erm(A)* and *aph(3′)-III*, where the reduction from 2015 to 2020 was significant, or *spc* and *erm(C)*, with a significant reduction between 2006 and 2020 (Figure 8). The significant reduction in *mecA*, *erm(A*), aph(3′)-III, and *spc* occurrence was observed mainly in isolates from Asia and North/South America, while in Africa also *erm(C)* showed a reduction in occurrence (Appendix A). In Asia, it was observed also a significant reduction in the occurrence of *dfrG* over the past two decades (Appendix A).

On the other hand, some ARGs increased their occurrence, such as *erm(B), fexA,* and *lnuB*, with a significant increase until 2015, and *tet(K)*, *tet(L)* and *tet(M)*, which increased significantly until 2010, with no significant differences thereafter (Figure 8 and Appendix A). However, in Asia, *tet(M)* showed a significant reduction over time.

The temporal analysis performed for global ARGs occurrence on plasmidic contigs also showed some significant differences (Figure 9 and Appendix A). The presence of *blaZ* increased significantly from 2001–2010 to 2011–2020, although a reduction was observed after 2015, while *erm(C)* occurrence decreased between 2010 and 2015, with a statistically significant difference between 2010 and 2020 (Figure 9). A significant decrease was also observed in *tet(K)* between 2005 and 2015 (Figure 9). Other genes that significantly decreased their prevalence were *erm(A)* and *spc*, between 2005 and 2020 (Figure 9). A similar reduction in *erm(C)*, *tet(K)*, *erm(A),* and *spc* was observed in isolates from Asia (Appendix A). An increase up to 2015 was observed for *tet(L)*, with a significant decrease thereafter, while *mph(C)* and *msr(A)* had a significant increase up to 2015, and *ant(6)-Ia* and *aph(3’)-III* also increased significantly (Figure 9 and Appendix A).

## 3. Discussion

In this study, we identified ARGs in the chromosomes and plasmidic contigs of publicly available genomes from *Staphylococcus aureus*. The performed analyses returned a global overview of the spread of ARGs and CCs worldwide according to isolation sources, and of temporal changes in ARGs occurrence. To our knowledge, this is the first study that includes *S. aureus* WGS data from worldwide, as previous studies were focused on specific regions or clonal complexes, for example CC5 in the western hemisphere or the strain USA300 (CC8) in USA [14,15]. Furthermore, the differentiation between plasmidic and chromosomic data provided a more precise perspective of the genomic location of ARGs. Obviously, this study has some limitations, firstly because of the lack of metadata in many genomes and even associated with biases due to predominance of genomic studies from different sources and/or countries, which could cause an under- or over-estimation in terms of geographical distribution, isolation sources, and temporal analysis. Furthermore, as new genomes are submitted continuously in the databases, these results represent a picture of the moment this study was undertaken, which can easily change considerably with the addition of new genomes to the analysis. Nevertheless, changes in ARG distribution during the last 20 years are consistently demonstrated.

Most of the isolates were MRSA (84% of human isolates were carrying *mecA*, against 74% of animal isolates), as the main ARG found along *S. aureus* genomes was the *mecA* gene, which was found only in chromosomes but not in the plasmidic contigs. It is well known that this gene, which confers resistance to beta-lactam antibiotics, is located on the Staphylococcal Cassette Chromosome mec (SCCmec), which can be exchanged between bacteria thanks to the presence of genes encoding the cassette chromosome recombinases (ccr) within the SCCmec [16]. On the other hand, the main ARG found in plasmidic contigs, the *blaZ* gene, which confers resistance to penicillin, was also found in some chromosomic contigs. These findings are in line with other research works, where the origin of genes associated with resistance to beta-lactams in staphylococci was attributed to transposon Tn552-like genes, while the *blaZ* gene was found on various mobile genetic elements (MGEs), such as transposons and plasmids. In our study, the overall presence of beta-lactam resistance genes was higher in the chromosomes than in the plasmidic contigs. Some other ARGs were found almost exclusively on plasmidic contigs, such as *ant(6)-Ia*, *aph(3′)-III*, *aac(6′)-aph(2″)*, *tet(M)*, and *dfrG*. The presence of these genes on MGEs has been documented in the past and their location on plasmids and other MGEs, such as transposons and integrons, has been previously described [17,18].

The geographical distribution analysis showed that some CCs are almost exclusively circumscribed to some continents, such as CC5 in North and South America; CC22 and CC398 in Europe; and CC8 in Africa, Oceania, and Asia. However, CC8 and CC30 were also abundant in Argentina. CC30 is nowadays the most common hospital and community acquired clonal complex in Argentina, which has replaced the main abundance of CC5 on the past decade [19]. The sequence type ST80 is considered the most abundant in Europe [20], but only 30 European isolates belonged to ST80 in our dataset, probably due to the lack of metadata about geographical location in the analysed public repositories. However, our study highlighted the presence of other CCs in European countries, for example, the high prevalence of CC398 in the Netherlands, followed by Germany, Italy, and Denmark, remarks the emerging spread of this clone in these areas. Furthermore, CC1 and CC59 are most prevalent in China, as has been found in previous studies [21,22,23], while CC8, CC45, and CC121 were the most abundant in Singapore, Taiwan, and Thailand, respectively. Other studies also described differences among Asian countries, for example, Turner et al. (2019) [24] identified CC8 as a common CC in Korea, CC30 in Japan, and CC59 in Taiwan, while Song et al. (2011) [25] described CC8 as widespread throughout Asian Countries and CC59 mostly isolated in Taiwan [24,25]. Our findings slightly differ from these examples, probably because of the different number of genomes considered due to a wider temporal interval and, again, a possible lack of metadata associated with the genomes available in public repositories.

The distribution of plasmids among CCs throughout the continents was similar to that of the chromosomes, however, the overall number of CCs with plasmid contigs associated was lower. It is well documented that plasmids are transferred between specific lineages in *S. aureus* and this could explain the differences in the abundance of plasmidic contigs belonging to different CCs [26,27]. Our results showed that plasmids associated with CC8 are widespread worldwide and this agrees with other studies highlighting the presence of plasmids in this CC [28,29,30]. Additionally, the presence of plasmids in other common CCs, such as CC398, CC22, CC5, and CC1, has also been documented in the past [26,31,32,33].

Our results showed that MRSA are still the most abundant *S. aureus* throughout the globe and that the genes conferring resistance to beta-lactams being both in chromosomes and plasmidic contigs. Chromosomes showing ARGs associated with resistance to aminoglycosides and trimethoprim and plasmidic contigs with ARGs for resistance to macrolides were found abundant in Asia. Liang et al. (2019) [34] showed that macrolide and aminoglycoside resistance genes were between the most represented on 103 MRSA clinical isolates from China, while *aac(6′)-aph(2″)*, *dfrG,* and *ermC* genes were also present with high occurrence [34]. The high prevalence of tetracycline resistance genes in Europe is in agreement with its association with CC398, which in this study was mainly found in European isolates. Tetracycline resistance is indeed a typical trait in CC398 isolates, as shown by previous studies [35,36].

The *dfrG* gene, which confers resistance to trimethoprim, was found significantly more abundant in Africa than in other continents, which is consistent with the antibiotic prescription and antimicrobial resistance data described in the literature [37,38]. On the other hand, the significant abundance of aminoglycosides ARGs in South America does not reflect the global use of this class of antibiotics, which seems to be more common in Asian countries, according to the data summarized on ResistanceMap [39]. However, ResistanceMap only includes data up to 2015 and on antibiotics used for human consumption [40], while the information of their use on farm animals is missing.

In our study, CC398 was found mainly in the Animal category, followed by Environment and Food categories. *S. aureus* CC398 was first isolated in humans, but it acquired the ability to also infect livestock, then humans in contact with livestock, and eventually to spread between humans [22,41,42].

Finally, CC5 and CC8 isolates were mainly associated with the human source, in concordance with outbreak and pandemic reports worldwide [43,44].

Our results showed the global distribution of ARGs in the different CCs analysed. In chromosomes, in almost all CCs, *mecA* was carried (absent only in CC130), reflecting the high incidence of MRSA. CC398 also harboured *erm(B)*, *tet(M)*, *tet(L)*, *dfrG*, *dfrK*, *erm(T)*, *fexA*, *lnuB,* and *str* genes, at higher levels compared to other CCs, similarly to the findings of previous works where isolates belonging to CC398 harboured resistance genes against aminoglycosides, beta-lactams, macrolides, tetracycline, and trimethoprim [36,45]. CC8 genomes showed high levels of *mecA*, *blaZ*, *tet(K)*, *aph(3′)-III*, *aac(6′)-aph(2″)*, *ant(6)-Ia,* and *tet(M)*. CC8 is known to carry resistance genes for beta-lactams, followed by a possible variety of other ARGs depending on the strain [46]. CC5 had high incidence of *mecA*, *blaZ*, *erm(A)*, *aadD,* and *spc* genes, which is in concordance with the findings by Challagundla et al. (2018) [14], where a phylogenomic study on CC5-MRSA found that the most common ARGs were those conferring resistance to beta-lactams, fluoroquinolones, macrolides, aminoglycosides, and tetracyclines [14]. Only *blaZ* was highly abundant in the chromosomes of CC30 isolates in our results, but *aadD*, *spc,* and *erm(A)* also had a high occurrence, especially in the plasmidic contigs, partially confirming the description of Moneke et al., where the most predominant strain, ST36/39-MRSA-II, carried SCCmec II with *aadD* and *erm(A)* in the plasmid pUB110 and the transposon Tn554, respectively [46].

In our study, CC130 carried on the chromosome only *mecC* and *str*, while *blaZ*, *erm(C)*, *tet(K)*, *tet(L),* and *str* genes were in the plasmidic contigs. In a study by Gómez et al. (2021) [23], only *mecC*, *blaZ,* and *tet(K)* were found as the main ARGs in CC130, showing that this CC carries ARGs mainly on plasmids [23].

The analysis of temporal changes highlighted an overall reduction along time in *mecA* and *erm(A)* on chromosomes, among other genes, while *blaZ* prevalence increased within plasmidic contigs. The same trend was found in a study on clinical isolates from two hospitals in USA during the period 2000 to 2014, where a decline of MRSA and erythromycin resistance was observed, while penicillin-resistant *S. aureus* increased [47]. Another study, analysing MRSA infections in patients of the Veterans Affairs medical centres in the USA from 2005 to 2017, also showed a decreasing trend of MRSA infections [48]. The increase in *blaZ* presence in plasmids, while it is decreasing in chromosomic regions, needs further investigations focused on the ecological advantages of keeping this gene in mobile regions, such as in the case of plasmids. Another significant trend was observed in the decrease over time of *erm(C)* occurrence, which is similar to the decline in the presence of MGEs carrying the *erm(C)* gene over the period 2001–2010 in a pool of 1193 *S. aureus* clinical isolates from Ireland [32]. Interestingly, *aph(3′)-III* had a significant reduction over time in the chromosomes and a significant increase in the plasmidic contigs in the past five years.

Many efforts have been made by the governments in the past to reduce the threat of antimicrobial resistance by monitoring and reducing the use of antibiotics in various sectors, following the One Health strategy [49,50,51,52,53,54,55,56,57]. The data gathered in this work can complement future works focused on understanding if the reductions observed in this study in the ARGs over time reflected the strategies adopted by countries for AMR control.

The present study has some limitations, which can be addressed by future research. First of all, it is important to highlight that low-income countries do not possess the same availability of technologies of high-income countries, which results in less data released from these countries, being the countries where AMR burden is higher.

Another limit is the necessity of constantly updated databases and software for the identification, for example, of all the different alleles of ARGs and different MGEs. This would bring much more detail into the analysis, identifying more ARGs on chromosomes and different MGEs, not only on plasmidic contigs. Furthermore, it would be interesting to expand the temporal analysis also to CCs, isolation sources, and geographical location, however, more effort is needed when submitting metadata related to the isolates to increase the statistical significance of the studies.

In conclusion, the aim of this study was to explore the resistome of *S. aureus* through analysing publicly available WGS data. The analysis, conducted on 29,679 *S. aureus* genomes, allowed to identify different ARGs, to locate them on chromosomes or plasmidic contigs and to associate them with CCs, isolation sources, geographical distribution, and temporal changes. Our results confirmed a global temporal reduction in MRSA and macrolide resistance occurrence and remarked on the increase in the occurrence of ARGs associated with aminoglycosides resistance in plasmids in recent years. The workflow used for this analysis is available for future resistome analyses on other species and can be adapted for more targeted outputs.

## 4. Materials and Methods

### 4.1. Download of Genomes and Associated Metadata

Assembled genomes and associated metadata were downloaded on February 2021 from NCBI [58], PATRIC [59], and PubMLST [60] databases, adapting the available Ruby scripts (https://github.com/JoseCoboDiaz/download_genomes, accessed on 12 November 2021). All the metadata from each database were adapted as previously described [61] and gathered in one table. Other categories were added manually to the metadata table, where possible, based on the subcategories present in the databases: the subcategory “collection_date” was grouped into “Date_Range”; a “Continent” was assigned manually where necessary, based on the “country” column; and “Isolation_Source” was grouped into “Category”.

### 4.2. Analysis of Genomes

The assembled genomes from each of the three databases were analysed with Staramr [62], which includes the ResFinder database [63] for ARGs identification. The scheme for MLST, from Pubmlst scheme [64], was also included in the Staramr pipeline. To automatise the analysis for the thousands of genomes available, the Ruby script 08.staramr_auto.rb (https://github.com/JoseCoboDiaz/download_genomes, accessed on 12 November 2021) was adapted and used. The clonal complexes (CCs) were assigned automatically, using the list of CCs associated with STs in the *S. aureus* PubMLST scheme [65].

Since some common CCs (CC398, CC130, CC59, and CC121) were missing in the PubMLST scheme, they were added manually into the list, based on the information gathered [66,67,68,69].

PlasFlow [70] was used for the identification of plasmidic contigs. A table with all the plasmidic contigs was created from the PlasFlow output files, which was used to split the ResFinder results into two files, one corresponding to chromosomic contigs and the other to plasmidic contigs. Two datasets including both resistome data and associated metadata were generated: one for chromosomic data and one for plasmidic data.

### 4.3. Data Analysis and Statistics

Both chromosomic and plasmidic datasets were explored and analysed in RStudio (Version 1.4.1106). Occurrence and patterns of distribution of AMR genes were analysed by CC, isolation source, temporal and geographical categories. For the temporal analysis, the isolates collected before 2001 were excluded due to the small sample size. Heatmaps were generated using the pheatmap R package. Only those genes present in at least 100 genomes were included for further analysis. Before performing the statistical analysis, the categories and subcategories were trimmed (categories consisting of < 100 genomes were excluded; subcategories of <10 genomes were excluded; all chromosomes or plasmids with missing metadata were also excluded). Normal distribution of occurrence of ARGs in the categories was assessed with the Shapiro–Wilk normality test and, based on the result, Student’s *t*-tests or Wilcoxon tests were performed to assess significant differences (*p* < 0.05) between each category of data. Boxplots were generated to visualise the results of the statistical analysis using the ggplot2 R package.

## Figures and Tables

**Figure 1 antibiotics-11-01632-f001:**
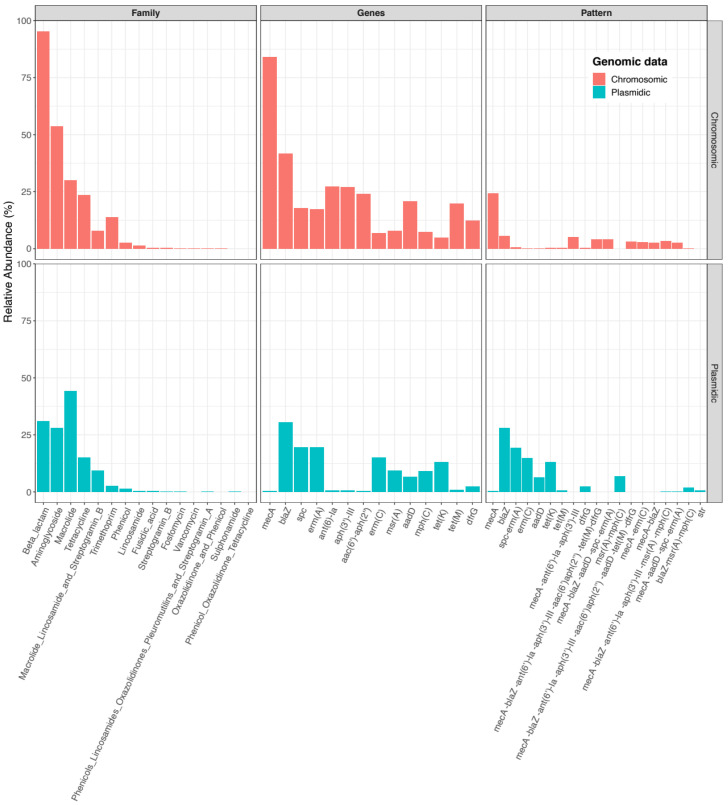
Global distribution of ARGs on *S. aureus* chromosomes and plasmidic contigs. ARG families, most abundant ARGs, and co-occurrence of ARGs were analysed.

**Figure 2 antibiotics-11-01632-f002:**
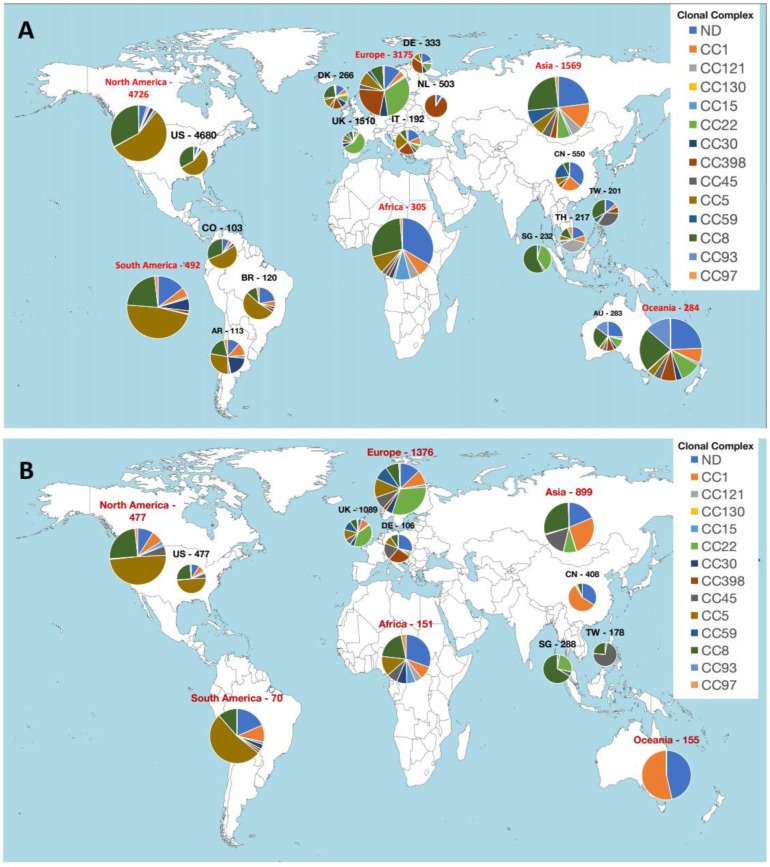
Geographical distribution of *S. aureus* clonal complexes in chromosomes (Chr) (**A**) and plasmidic contigs (PC) (**B**). Except for South America, only countries with more than 100 Chr or PC were represented. The number indicates the Chr or PC associated with the corresponding continent or country. Colours within pie charts represent the percentage of Chr/PC belonging to the main CCs. Genomes with unassigned CCs are marked as “ND”. Country abbreviations: AR, Argentina; AU, Australia; BR, Brazil; CO, Colombia; CN, China; DE, Germany; DK, Denmark; IT, Italy; SG, Singapore; TH, Thailand; TW, Taiwan; UK, United Kingdom; US, United States of America.

**Figure 3 antibiotics-11-01632-f003:**
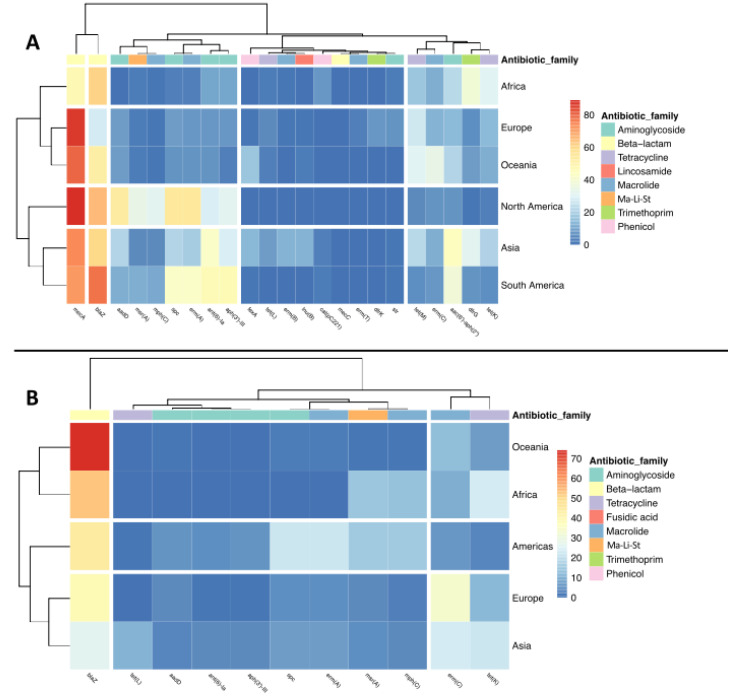
Heatmaps showing the abundance of ARGs in chromosomic (Chr) (**A**) and plasmidic contigs (PC) (**B**) in each continent. Only ARGs present in more than 100 Chr or 80 PC were included. In PC, North and South America were pulled together to increase the sample size. Ma-Li-St: macrolide, lincosamide, and streptogramin B.

**Figure 4 antibiotics-11-01632-f004:**
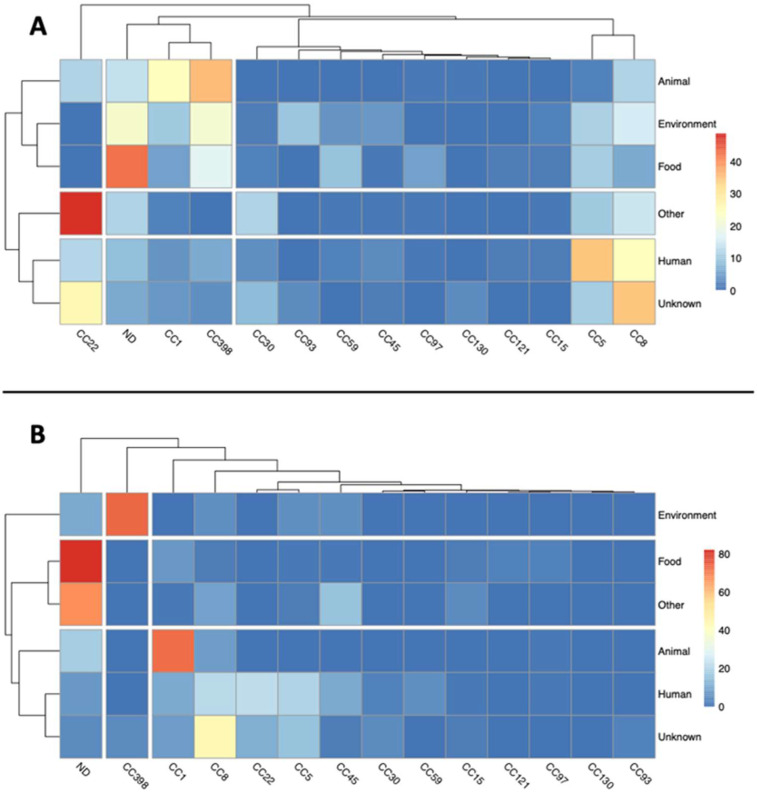
Heatmaps showing the CC distribution throughout the isolation source categories in chromosomes (Chr) (**A**) and plasmidic contigs (PC) (**B**). The category “Other” includes all the isolation sources that could not be clearly assigned to the rest of categories.

**Figure 5 antibiotics-11-01632-f005:**
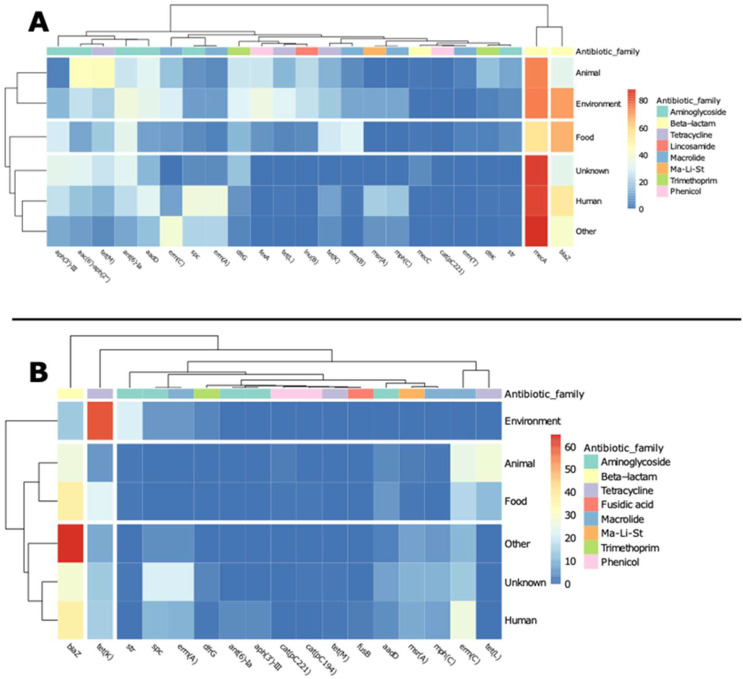
Heatmaps showing the ARGs distribution through the isolation sources in chromosomes (Chr) (**A**) and plasmidic contigs (PC) (**B**). The source “Other” includes all the isolation sources that could not be clearly assigned to the rest of categories. Ma-Li-St: macrolide, lincosamide, and streptogramin B.

**Figure 6 antibiotics-11-01632-f006:**
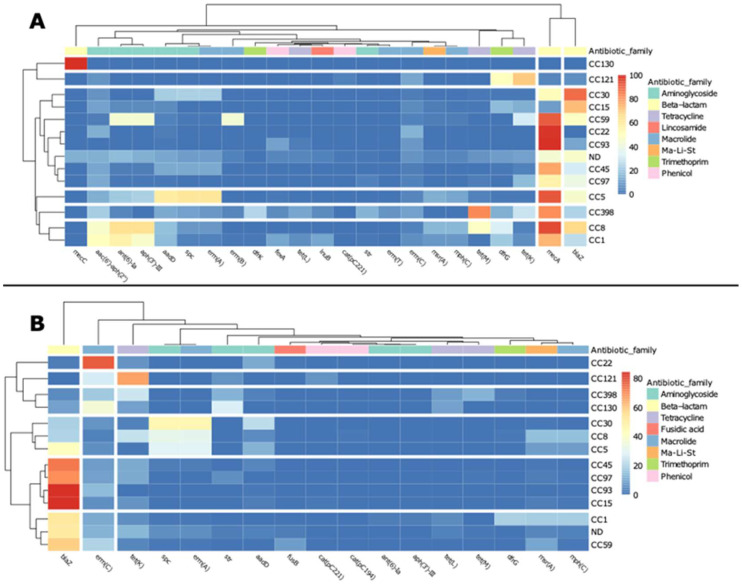
Heatmaps showing the ARGs distribution among *S. aureus* CCs in chromosomes (Chr) (**A**) and plasmidic contigs (PC) (**B**). Heatmap plot showing the percentage of genomes per CC associated with resistance to antibiotics. Ma-Li-St: macrolide, lincosamide, and streptogramin B.

**Figure 7 antibiotics-11-01632-f007:**
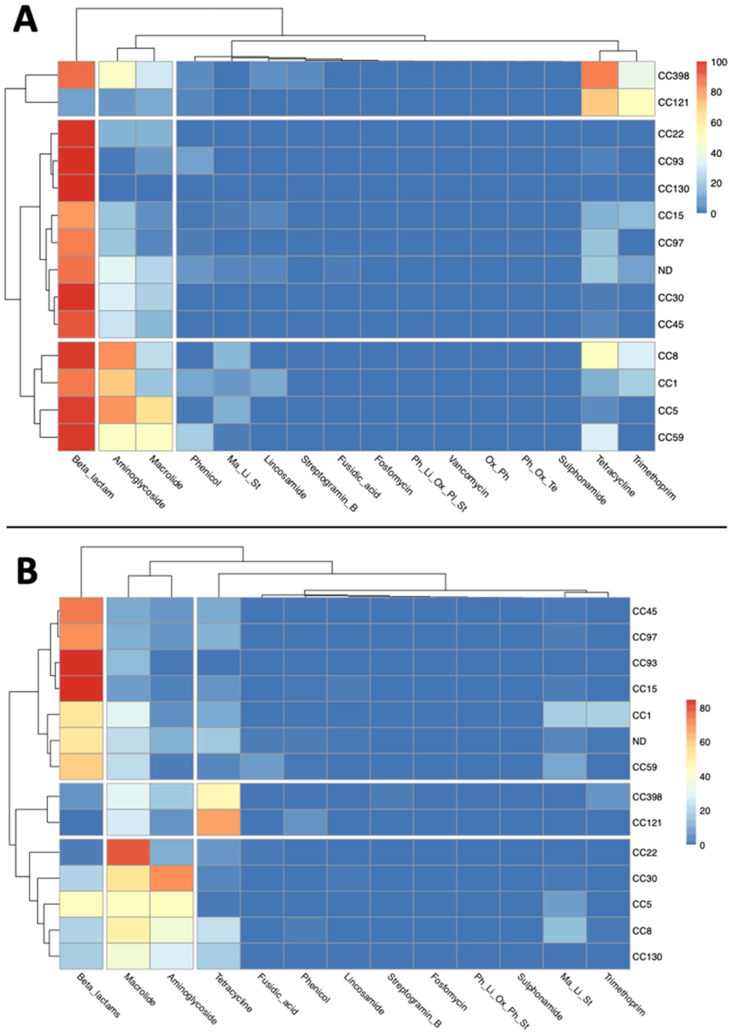
Heatmaps showing the ARGs distribution among *S. aureus* CCs in chromosomes (Chr) (**A**) and plasmidic contigs (PC) (**B**). Heatmap plot showing the percentage of genomes per CC carrying ARGs associated with resistance to different antibiotic families. Ma_Li_St: macrolide, lincosamide, and streptogramin_B; Ph_Li_Ox_Pl_St: phenicols, lincosamides, oxazolidinones, pleuromutilins, and streptogramin_A; Ox_Ph: oxazolidinone and phenicol; Ph_Ox_Te: phenicol, oxazolidinone, and tetracycline.

**Figure 8 antibiotics-11-01632-f008:**
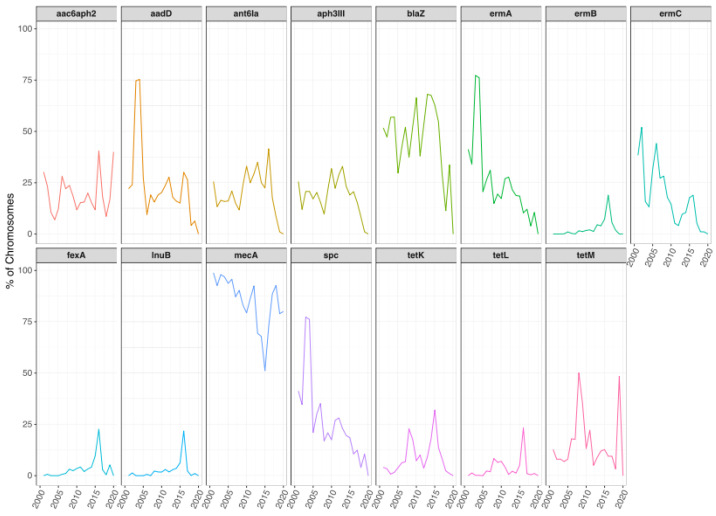
Temporal changes in the resistome of *S. aureus* chromosomes. Line plots showing the trends of occurrence for each ARG during the years. Only ARGs that showed significant differences are shown.

**Figure 9 antibiotics-11-01632-f009:**
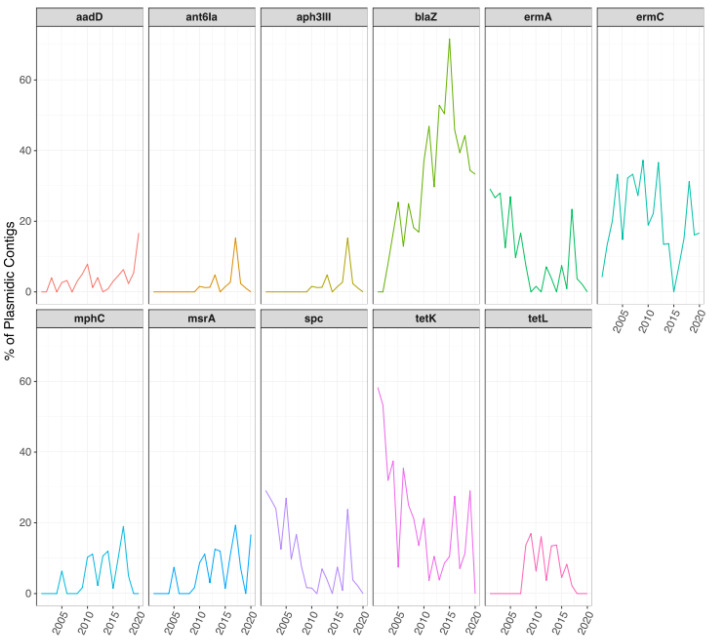
Temporal changes in the resistome of *S. aureus* plasmidic contigs. Line plots showing the trends of occurrence for each ARG during the years. Only ARGs that showed significant differences are shown.

**Table 1 antibiotics-11-01632-t001:** Most abundant antimicrobial resistance genotypes, arranged by decreasing relative abundance. Only genotypes with relative abundance >1% are shown.

# Pattern	# Contigs	% Contigs	% Cumulative	# ARGs	ARG Genotype—Plasmidic Contigs
**1**	5862	27.91	27.91	1	*blaZ*
**2**	4068	19.37	47.27	2	*spc—erm(A)*
**3**	3115	14.83	62.10	1	*erm(C)*
**4**	2741	13.05	75.15	1	*tet(K)*
**5**	1420	6.76	81.91	2	*msr(A)—mph(C)*
**6**	1322	6.29	88.20	1	*aadD*
**7**	491	2.34	90.54	1	*dfrG*
**8**	392	1.87	92.41	3	*blaZ—msr(A)—mph(C)*
**# Pattern**	**# Chromosomes**	**% Chromosomes**	**% Cumulative**	**# ARGs**	**ARG Genotype—Chromosomes**
**1**	6002	24.24	24.24	1	*mecA*
**2**	1416	5.72	29.95	1	*blaZ*
**3**	1233	4.98	34.93	3	*mecA—ant(6)-Ia—aph(3’)-III*
**4**	1046	4.22	39.16	7	*mecA—blaZ—ant(6)-Ia—aph(3’)-III—aac(6’)-aph(2’’)—tet(M)—dfrG*
**5**	1031	4.16	43.32	5	*mecA—blaZ—aadD—spc—erm(A)*
**6**	831	3.36	46.67	6	*mecA—blaZ—ant(6)-Ia—aph(3’)-III—msr(A)—mph(C)*
**7**	764	3.08	49.76	8	*mecA—blaZ—ant(6)-Ia—aph(3’)-III—aac(6’)-aph(2’’)—aadD—tet(M)—dfrG*
**8**	726	2.93	52.69	2	*mecA—erm(C)*
**9**	676	2.73	55.42	4	*mecA—aadD—spc—erm(A)*
**10**	668	2.70	58.12	2	*mecA—blaZ*
**11**	628	2.54	60.65	2	*mecA—aac(6’)-aph(2’’)*
**12**	608	2.46	63.11	1	*mecC*
**13**	459	1.85	64.96	2	*mecA—aadD*
**14**	324	1.31	66.27	4	*mecA—blaZ—aac(6’)-aph(2’’)—tet(M)*
**15**	302	1.22	67.49	4	*mecA—ant(6)-Ia—aph(3’)-III—aac(6’)-aph(2’’)*

## Data Availability

Data is available in the Appendix A.

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
