# Peer review of "Antimicrobial Resistance Genes Analysis of Publicly Available Staphylococcus aureus Genomes"

_antibiotics, 2022, doi:10.3390/antibiotics11111632_

Round 1

Reviewer 1 Report

The manuscript Pennone et al., “Antimicrobial resistance genes analysis of publicly available Staphylococcus aureus genomesreports on a study of the ESCAPE pathogen resistome through analysis of publicly available whole genome sequencing data. The analysis, conducted on almost 30000 S. aureus genomes, allowed to identify and allocate 13 different clonal complexes (CC) on chromosomes or plasmids identifying 73 and 47 antibiotic resistance genes (ARGs) respectively. Associating ARGs with, isolation sources, geographical distribution and temporal changes showing reduction in Methicillin-resistant Staphylococcus aureus and the increase in the occurrence of 23 ARGs associated with aminoglycosides resistance.

1.     While overall research design and implementation pipeline is appropriate, the sufficiency of metadata is not obvious thus additional information on metadata used in form of Table in main text would be recommended.

2.     As section Results in current version of the manuscript is largely plain reflection of data shown in figures and tables and thus is somewhat monotonous, this section might be combined with Discussion.

3.     Also understanding the difficulty of such a task, additionally, it would be highly relevant to compliment the obtained results with data available on antibiotic usage against S. aureus within last decades.

4.     More in depth analysis (at least hypothesis) why the reduction (also statistically insignificant) in blaZ presence in chromosomes is occurring in parallel with significant increase on plasmidic contigs would additionally improve the manuscript.

Overall, this work might serve a great interest not only for the scientists working with S. aureus, but also might give valuable insight of antibiotic resistance global prevalence and evolution for epidemiologists and medical society in general.

Specific comments

·      Tables and Figures: Please increase the resolution of axis titles.

·      S. aureus  should be always in Italic.

·      Spelling and grammar mistakes can be found.

Author Response

1. While overall research design and implementation pipeline is appropriate, the sufficiency of metadata is not obvious thus additional information on metadata used in form of Table in main text would be recommended.

A resume of the obtained metadata information is available on Supplementary tables 1 and 2. Moreover, supplementary tables 3 and 4 present the metadata information for each chromosome and plamid, respectively, manually curated from those obtained in the corresponding repositories at columns 12 to 18 (or R to L In excel or from “date range” to “Category”). Unfortunately, the lack of information (high percent of “unknown”) is very common on such genomes repositories.
To clarify the presence of such metadata in the supplementary information, “(Supplementary tables S1 and S2,
metadata information per chromosomic or plasmidic contig can be found at Supplementary tables S3 and S4)” has been added to line 93.

2. As section Results in current version of the manuscript is largely plain reflection of data shown in figures and tables and thus is somewhat monotonous, this section might be combined with Discussion.

The authors understand your proposal but we would prefer to keep the two sections separated, as the rest of reviewers are agree with the format employed.

3. Also understanding the difficulty of such a task, additionally, it would be highly relevant to compliment the obtained results with data available on antibiotic usage against S. aureus within last decades.

The main problem to address this information is the difficulty to obtain such information for each country analyzed, so we indicate briefly this point at lines 476-482.

4. More in depth analysis (at least hypothesis) why the reduction (also statistically insignificant) in blaZ presence in chromosomes is occurring in parallel with significant increase on plasmidic contigs would additionally improve the manuscript.

The sentence “he increase in blaZ presence in plasmids, while it is decreading in chromosomic regions, needs further investigations focusedon ecological advantages of keep this gene mobile regions such as plasmids.” has been added at lines L522-524.

Reviewer 2 Report

The article by Pennone et al. examines the frequency of antimicrobial resistance genes in Staphylococcus aureus from available whole genome sequences. Analysis examines both chromosomal genes as well as genes located on plasmids. Changes in frequency of antibiotic resistance genes over time was also determined. The study finds distinct geographical localizations of specific clonal clusters when data is available as well as a global decline in MRSA and macrolide resistance, but an increase in aminoglycoside resistance. Overall the article is well written and the use of such a large dataset lends additional validity to the findings. There are some minor issues that should be addressed and are detailed below.

Major:

The available metadata gave geographic information for 43% of the genomes (ln 82). It is unclear what genomes were analyzed in the text. For analysis that is specified for geographic locations, clonal clusters and resistance allele frequency, the strains used had the associated metadata. When analysis then examines temporal variations of resistance genes it is unclear what genomes are being analyzed. The entire dataset downloaded, the genomes that were included in the geographical analysis, or chromosomes that may have date of isolation but not location. The use of 30,000 isolates vs 13,000 (43% of downloaded isolates) for global changes in allele frequency is a large difference. Clarification is required as to the number of strains actually used for analysis.

Minor:

Throughout the text S. aureus needs to be italicized.

Throughout the text gene names are not italicized and this should be corrected.

Figure 3: Text is hard to read in the figure.

Figure 5: Text is hard to read in the figure.

Figure 6: Text is hard to read in the figure.

Author Response

The available metadata gave geographic information for 43% of the genomes (ln 82). It is unclear what genomes were analyzed in the text. For analysis that is specified for geographic locations, clonal clusters and resistance allele frequency, the strains used had the associated metadata. When analysis then examines temporal variations of resistance genes it is unclear what genomes are being analyzed. The entire dataset downloaded, the genomes that were included in the geographical analysis, or chromosomes that may have date of isolation but not location. The use of 30,000 isolates vs 13,000 (43% of downloaded isolates) for global changes in allele frequency is a large difference. Clarification is required as to the number of strains actually used for analysis.

The sentence “From the whole dataset, information regarding the isolation dates (from 2001 to 2020) were present only in 10927 chromosomes (44%) and 2855 plasmidic contigs (14%), which were employed for following analysis.” has been added at lines 338-340.

Minor:

Throughout the text S. aureus needs to be italicized.

The proposed change has been done along the text. Sorry for this issue and thanks to appoint it.

Throughout the text gene names are not italicized and this should be corrected.

The proposed change has been done along the text. Sorry for this issue and thanks to appoint it.

Figure 3: Text is hard to read in the figure.

Figure 5: Text is hard to read in the figure.

Figure 6: Text is hard to read in the figure.

Text in figures 3, 5 and 6 have been changed to address the issues proposed by the reviewers.

Reviewer 3 Report

General comments

The authors present results of bioinformatic analysis on chromosomal and plasmidic AMR genes and clonal groups of a huge number of globally deposited Staphylococcus aureus genomes (from between 2001-2020), regardless of their distribution of MRSA or non-MRSA. The results confirm the trends of an overall decrease of MRSA (indicated by the presence of mecA gene and spc genes ?) and of macrolide resistance. On the other hand they report an overall increasing trend of plasmid associated aminoglycoside genes and beta-lactamase resistance gene blaZ (except between 2015-2020 for blaZ), and concluding that the latter resistance genes remain the most abundant on S. aureus throughout the globe. Results also provide an overview on geographical distribution of clonal complexes (CCs), largely confirming earlier reports on characteristic CCs of different continents.

This treasury of information is representing a great value of this work but the limitations – mentioned in text by the authors – should also be taken into account when interpreting the representative value of the results. Although the limitations such are inherent of this collection, it should be kept in mind that due to missing metadata, information on the sample source was not available for 52% of chromosomal and for 86% of plasmidic contigs, and that the number of isolates from different sources of differing continents were not balanced enough. Furthermore that 57% of the chromosomal and 85% of the plasmidic contigs were from unknown countries.

Specific Comments

1.      The Abstract should give the time period (2001-2020) of isolate data. Furthermore there should be at least one sentence also in the Abstract about the limitations for interpretation the results (see missing information on sources and on countries of origin).

2.      Also in the Abstract (L20) should be pointed out that blaZ was also abundant (42%) on chromosomes. This would bring the harmony with the statement about overall presence of blaZ higher in the chromosomes than in the plasmidic contigs (L347348).

3.      Results: Fig.3. should have the AMR gene names more clearly printed (like they are on Fig.1).

4.      In Results on temporal analysis of plasmidic blaZ there seems to be a contradiction between text and Fig 9 (on blaZ). Fig.9 shows an increasing trend for blaZ, but except between 2015-2020. The text (L306) says: “blaZ increased significantly from 2001-2010 and 2011-2020”.

5.      Supplementary Table 11 (ST11) is mentioned in the text as on “choromsomal contigs” (L510-511) although this table is about plasmidic contigs.

6.      Regarding the outstanding clinical significance of MRSA, it is suggested that in the Results section (or at least in the Discussion) a paragraph should be devoted to an analysis of these data regarding the overall differences between MRSA and non-MRSA isolates in terms of AMRGs and clonal complexes. Otherwise there should be a reason given why this distintion has not been taken in the study but MRSA was suddenly brought into the conclusion (L455) and in to the Abstract (L23).

Author Response

1. The Abstract should give the time period (2001-2020) of isolate data. Furthermore, there should be at least one sentence also in the Abstract about the limitations for interpretation the results (see missing information on sources and on countries of origin).

Sentence “(from 2001 to 2020) ” has been added at line 27; and “Despite the lack of metadata information in around half of the genomes analyzed, theresults obtained allows” at lines 30-31, to address the proposed changes.

2. Also in the Abstract (L20) should be pointed out that blaZ was also abundant (42%) on chromosomes. This would bring the harmony with the statement about overall presence of blaZ higher in the chromosomes than in the plasmidic contigs (L347348).

To address this issue, the sentence “was the most abundant in plasmidic contigs (30%), although it was also abundaant in chromosomes (42%).” was added at line 25.

3. Results: Fig.3. should have the AMR gene names more clearly printed (like they are on Fig.1).

Text in figures 3, 5 and 6 have been changed to address the issues proposed by the reviewers.

4. In Results on temporal analysis of plasmidic blaZ there seems to be a contradiction between text and Fig 9 (on blaZ). Fig.9 shows an increasing trend for blaZ, but except between 2015-2020. The text (L306) says: “blaZ increased significantly from 2001-2010 and 2011-2020”.

The significant difference was found between the two time-frames of 2001-2010 and 2011-2020, despite the reduction observed after 2015. The sentence “although a reduction was observed after 2015, ” has been adedd at line 369 to clarify this point.

5. Supplementary Table 11 (ST11) is mentioned in the text as on “chromosomal contigs” (L510-511) although this table is about plasmidic contigs.

Issue corrected at line 625.

6. Regarding the outstanding clinical significance of MRSA, it is suggested that in the Results section (or at least in the Discussion) a paragraph should be devoted to an analysis of these data regarding the overall differences between MRSA and non-MRSA isolates in terms of AMRGs and clonal complexes. Otherwise, there should be a reason given why this distinction has not been taken in the study but MRSA was suddenly brought into the conclusion (L455) and in to the Abstract (L23).

More relevance has been given in the Discussion section regarding MRSA, by adding sentence “The most of the isolates were MRSA (84% of human isolates were carrying mecA, against 74% of animal isolates), as” at line 404; “MRSA are still the most abundant S. aureus throughout the globe and that ” at line 459; and “ almost in all CCs mecA was carried (absent only in CC130), reflecting the high incidence of MRSA.” at line 491.

Reviewer 4 Report

Pennone et al. have done a nice piece of work and it is a current need to analyze the existing genomic data for future research planning. However, the current study needs some minor revisions before the acceptance of the paper.

Abstract

1.     Line 18, 20: Bacterial scientific names and resistant genes should be italicized

2.     Line 22-23: Please rephrase in an understandable manner to reader.

Introduction

1.     Line 43: S. aureus should be italicized and please correct this mistake that appear through out the manuscript.

2.     The author has mentioned a list of favorable points on using WGS in bacterial typing. That’s interesting. I recommend to add the limitations of using WGS and how less practical to use WGS in resource poor settings. Because, nowadays, highest AMR burden can be seen in resource poor countries.

Results

1.     All the resistant genes should be italicized throughout the manuscript

2.     Line 162-174: could this continental AMR genes distribution match with countries antimicrobial usage rates https://www.who.int/initiatives/glass

3.     What are the findings for currently available antimicrobials (vancomycin, fluoroquinolones or minocycline) resistance among the considered genomes. This is important to address as well.

4.     What is your finding specifically on animal origin MRSA as this is important in One-health perspective

Discussion

1.     Please explain the translation of data you have generated in the current study to AMR mitigation policy development and the possible future research directions as the whole purpose of the study needs to address

Methods

1.     Please mention the reference genomes used inn the study

Others

1.     Please add the answers to line 524-527  

Author Response

Abstract

1. Line 18, 20: Bacterial scientific names and resistant genes should be italicized

The proposed change has been done.

2. Line 22-23: Please rephrase in an understandable manner to reader.

The proposed change has been done: “showed that, in plasmids, MRSA and macrolide resistance occurrence decreased, while the occurrence of ARGs associated with aminoglycosides resistance increased”

Introduction

1. Line 43: S. aureus should be italicized and please correct this mistake that appear through out the manuscript.

The proposed change has been done.

2. The author has mentioned a list of favorable points on using WGS in bacterial typing. That’s interesting. I recommend to add the limitations of using WGS and how less practical to use WGS in resource poor settings. Because, nowadays, highest AMR burden can be seen in resource poor countries.

The sentence “it is important to highlight that low-income Countries do not possess the same availability of technologies of high-income Countries, which results in less data released from these countries, being the countries where AMR burden is higher.” has been added at lines 541-544.

Results

1. All the resistant genes should be italicized throughout the manuscript

The proposed change has been done.

2. Line 162-174: could this continental AMR genes distribution match with countries antimicrobial usage rates https://www.who.int/initiatives/glass

This is a very important point and in the Discussion section it has been pointed out at lines 476-482. However, GLASS surveillance system reports the use of antibiotics to treat human cases (cefoxitin or oxacillin), while in this paper we are considering the global distribution (not only in human) of S. aureus and its ARGs.

3. What are the findings for currently available antimicrobials (vancomycin, fluoroquinolones or minocycline) resistance among the considered genomes. This is important to address as well.

ARGs against these antimicrobials were not detected or their occurrence was too low to be included as the most important ARGs in the analysis (information for all ARGsis available in Supplementary Tables 3 and 4)

4. What is your finding specifically on animal origin MRSA as this is important in One-health perspective

This was briefly indicated at lines483-486.

Discussion

1. Please explain the translation of data you have generated in the current study to AMR mitigation policy development and the possible future research directions as the whole purpose of the study needs to address

Sentence “The data gathered in this work can complement future works focused on understanding if the reductions observed in this study in the ARGs in the Continents over time reflected the strategies adopted by the Countries for AMR control.” added at lines 535-538.

Methods

1. Please mention the reference genomes used in the study

No reference genomes were used in this study

Others

1. Please add the answers to line 524-527

We don’t understand this proposed change.